# Enhanced Ibuprofen Adsorption and Desorption on Synthesized Functionalized Magnetic Multiwall Carbon Nanotubes from Aqueous Solution

**DOI:** 10.3390/ma13153329

**Published:** 2020-07-27

**Authors:** Ghadir Hanbali, Shehdeh Jodeh, Othman Hamed, Roland Bol, Bayan Khalaf, Asma Qdemat, Subhi Samhan

**Affiliations:** 1Department of Chemistry, Faculty of Science, An-Najah National University, P. O. Box 7, Nablus 00001, Palestine; g.hanbali@najah.edu (G.H.); bayan.kh107@hotmail.com (B.K.); 2Institute of Bio and Geosciences, Agrosphere (IBG-3), Forschungszentrum Jülich GmbH, 52425 Jülich, Germany; r.bol@fz-juelich.de; 3Jülich Center for Neutron Science and Peter Grunberg Institute, Forschungszentrum Jülich GmbH, 52425 Jülich, Germany; qdemat@fz-juelich.de; 4Palestinian Water Authority, Ramallah 00001, Palestine; subhisamhan@yahoo.com

**Keywords:** magnetic multi-wall carbon nanotube, adsorption, kinetics, isotherm, ibuprofen

## Abstract

In recent years, concerns have been raised about the occurrence of active raw materials and pharmaceutical ingredients that may be present in water, including wastewater, in the pharmaceutical industry. Wastewater treatment methods are not enough to completely remove active pharmaceuticals and other waste; thus, this study aims to assess the use of a multiwall carbon nanotube after derivatization and magnetization as a new and renewable absorbent for removing ibuprofen from an aqueous medium. The adsorbents were prepared by first oxidizing a multiwall carbon nanotube and then deriving the oxidized product with hydroxyl amine (m-MWCNT–HA), hydrazine (m-MWCNT–HYD), and amino acid (m-MWCNT–CYS). Adsorbents were characterized by Raman spectroscopy, Fourier Transform infrared spectroscopy (FTIR), scanning electron microscopy (SEM and TEM), Brunauer–Emmett–Teller surface area analysis (BET), thermogravimetric analysis (TGA), and vibrating sample magnetometer (VSM). Batch adsorption studies were conducted to study the effects of pH, temperature, time, and initial concentration of the adsorbate. Adsorption isotherm, kinetics, and thermodynamics studies were also conducted. The results show that the optimal pH for nearly complete removal of Ibu in a short time at room temperature was 4 for three adsorbents. The adsorption followed the Langmuir isotherm model with pseudo-second-order kinetics. The percentage of removal of ibuprofen reached up to 98.4%, 93%, and 61.5% for m-MWCNT–CYS, m-MWCNT–HYD, and m-MWCNT–HA respectively. To the best of our knowledge, the grafted MWCNTs presented in this work comprise the first example in the literature of oxidized MWCNT modified with such functionalities and applied for ibuprofen removal.

## 1. Introduction

Pharmaceuticals are substances used in the diagnosis, treatment, or prevention of disease and for restoring, correcting, or modifying organic functions (Encyclopedia Britannica), whereas personal care products (PCPs) are primarily used to improve the quality of life [1].

In recent years, concerns have been raised about the occurrence of active raw materials and pharmaceutical ingredients that may be present in water, including wastewater, in the pharmaceutical industry.

Pharmaceuticals may enter the environment through an incalculable number of scattered points and thus can be categorized into different types: hormones, anti-epileptics, anti-inflammatory, anti-depressants, beta-blockers, statins, antibiotics, etc. [2].

Due to their continuous potential to inflict physiological impacts in humans at low doses, pharmaceuticals and Personal Care Products (PPCPs) are considered a unique set of emerging environmental pollutants.

The broad nature of the use of global chemicals, combined with the advent of new pharmaceuticals on the market, significantly contributes to the presence of these chemicals in the aquatic environment [3].

However, in some wastewaters, despite their having been treated by wastewater treatment plants (WTP), there are certain pharmaceuticals and other contaminants that couldn’t be treated for. These treated wastewaters are almost always used for agricultural purposes [4]. Multiwall carbon nanotube (MWCNT) is being employed in a large number of commercial applications due to its excellent properties, such as tensile strength, high mechanical strength, high conductivity when correctly joined in a composite structure, thermal stability at over 600 °C, and large specific surface area [5]

A common method is to enhance the dispersion, and optimize the use of multi-wall carbon nanotubes through chemical functionality. This enables the formation of chemical interconnection between MWCNT and targeted materials. Both non-covalent and covalent structures have been employed to improve solubility and chemical modifications to MWCNT. Overall, the interconnection of molecules to MWCNT by covalent bonds is more stable and effective due to the fact that grafted molecules increase MWCNT solubility, even with a low degree of functionality [6].

Many reports have confirmed that MWCNTs are used for water treatment and heavy metal adsorption, for example, for removal of Zn^2+^, Cu^2 +^, Cd^2+^, and Pb^2+^. Carbon nanotubes can also be used to extract polar organic molecules from water. This is possible due to the interactions in the form of hydrophobic effects, π–π interactions, covalent bonding, hydrogen bonding [7] and electrostatic interactions [8].

In recent years, there has been a focus on magnetic adsorbents in order to prevent issues linked to adsorbent regeneration. The key advantages of magnetic composites include: high performance, rapid adsorption rate, and increased adsorption ability [9]. The composite adsorbent could effectively accomplish the solid–liquid separation under an active magnetic field with no need for filtration or centrifugation. This is a result of the magnetism properties, and simplifies post-processing. In fact, magnetic recycling may help avoid the occurrence of nano-adsorbents in the natural environment, and minimize future hazards [10]. Ibuprofen (Ibu) is a non-steroidal anti-inflammatory drug, it has been found to exist in sewer wastewater in significant quantities up to 10 µg/L and 169 µg/L [11]. The presence of Ibu in nature can have traumatic effects on living organisms. The combination of Ibu with other drugs can stop cell reproduction in human embryos, and some reports show that contact with Ibu can have a negative effect on aquatic vertebrate reproduction [12]. Unfortunately, sewage treatment plants (STPs) are not really effective in the treatment of such wastewater, because they are not designed to remove pharmaceuticals [13]. Therefore, several other technologies have been reported for the removal of Ibu from contaminated water; among these are adsorption, membrane filtration, photodegradation, etc. [14,15,16,17,18,19,20].

With reference to previous studies, Bakr and Rahaman [14] employed membrane filtration for Ibu removal and achieved 100% results. Wang et al. [20] showed that PVDF membrane on multi-walled carbon nanotubes was able to remove 27%, but this is considered a high-maintenance technique; additionally, operation costs are high, and high-energy requirements are often essential for the technique to work. The design of membrane filtration systems can differ significantly, and rapid membrane clogging can result in big problems. The choice of the membrane is application-specific (hardness reduction, particulate or total organic carbon). Yuan et al. [15] was able to remove 89% of Ibu using photodegradation by TiO_2_ co-doping with urea and functionalized carbon nanotube (CNT). Zhao et al. [17] was able to remove 47% of Ibu using photodegradation via solar UV lamp, but this method has several drawbacks; photodegradation is often accompanied by formation of dioxins and others pollutants (metals, etc.), and it is also often very time consuming. Adsorption have been proved to be a cost-effective method for the removal of several pollutants from wastewater. In addition, the sorbent can be regenerated and reused for many times. Oyetade et al. [19] reported that maximum adsorption of Ibu was 12.2 mg/g using carboxylated carbon nanotube (CNT–COOH). Banerjee et al. [21] removed Ibu from aqueous solutions using graphene oxide nanoplatelets (GONPs), and the results showed that the degradation reached up to 98.17%. Bhadra et al. [22] used three adsorbents to remove Ibu; activated carbon (AC) was the first one. The second was a Zeolitic-imidazolate framework (Zif 8), and finally, the third was porous carbons derived from MOF (PCDM 1000). The adsorption capacity of PCDM 1000 is 320 mg/g, which is about three times greater than that of AC and about 11.4 times greater than that of Zif 8. Hydrogen bonds may explain the principal adsorption mechanism between PCDM 1000 and Ibu. 

There are several mechanisms of Ibu adsorption on functionalized MWCNT, such as π–π interaction, hydrophobic interaction, hydrogen bonding, π–hydrogen bonding and Lewis acid–base interaction [23].

This work focused on an MWCNT grafted with various functionalities as a potential new adsorbent for Ibu. The MWCNT was first oxidized, then converted to acid chloride. MWCNT functionalized with acid chloride was reacted separately with hydroxylamine, cystine, and hydrazine. The prepared grafted MWCNTs were magnetized for easier separation and regeneration, and evaluated as an adsorbent for Ibu from water and sample. The effect of various factors on the adsorption efficiencies of grafted MWCNT were evaluated. All results were characterized with different analytical techniques like SEM, TEM, FT-IR, TGA, BET and VSM.

The novelty of this work can be summarized as being that, to the best of our knowledge, the grafted MWCNTs presented in this work comprise the first example in the literature of oxidized MWCNT modified with such functionalities and applied for ibuprofen removal. Additionally, previous studies on the removal of Ibu using a single functional group provided lower removal than that presented in this study. Other studies with higher removals than that presented in this study were based on using photodegradation or membranes, which are extremely expensive methods compared with the method presented in our study.

## 2. Materials and Methods

### 2.1. Chemicals and Materials

All chemicals, including nitric acid (HNO_3_), hydrogen peroxide (H_2_O_2)_, oxalyl dichloride(C_2_O_2_Cl_2)_, triethylamine (TEA, Et3N), dimethylformamide (DMF, (CH_3_)_2_NC(O)H), tetrahydrofuran (THF, (CH_2_)_4_O), hydrochloric acid (HCl), sodium hydroxide (NaOH), ammonium hydroxide (NH_4_OH), pyridine (C_5_H_5_N), hydroxylamine (NH_2_OH), hydrazine(N_2_H_4_), cystine (C_6_H_12_N_2_O_4_S_2)_, ferric Chloride hexahydrate (FeCl_3_·6H_2_O), ferrous chloride tetrahydrate (FeCl_2_·4H_2_O) were of analytical grade. All chemicals used in this work were purchased from Sigma-Aldrich (Jerusalem, Israel) and used as received. MWCNT (purity ≥ 99.5%) was purchased from Sigma-Aldrich (Amman, Jordan). Ibuprofen standard was of high purity grade (purity ≥ 99%) and was purchased from Jerusalem Pharmaceuticals Company, Ramallah, Palestine. HPLC grade water, acetonitrile and methanol, distilled water was also used to prepare stock solutions.

### 2.2. Preparation of Oxidized Multiwall Carbon Nanotube (MWCNT–COOH)

In this step, 0.1 g of MWCNT was treated with 100 mL of 69% HNO_3_ in a flask of 500 mL. The flask was vibrated in an ultrasonic bath for 30 min at 25 °C. Next, the mixture was diluted with deionized water to reach 400 mL and then filtered through a polycarbonate membrane (0.22 μm). The same procedure was repeated exactly with H_2_O_2_ (30% v/v) instead of HNO_3_. Hydrogen peroxide was used to complete the oxidation process but mildly. The pH of the filtrate was roughly 7.0 by washing the solid with deionized water, then the product was dried under a 24 h vacuum to produce the carboxylic acid-functionalized MWCNT(MWCNT–COOH) [24].

### 2.3. Preparation of Acylated Multiwall Carbon Nanotube

Oxidized MWCNTs (MWCNTs–COOH), (0.1 g) were stirred in 2 mL of oxalyl chloride in the presence of 2–3 drops of dimethylformamide (DMF) and 2 mL of triethyl amine (TEA) at 70 °C for 24 h under N_2_. After cooling down to room temperature, the excess TEA was washed repeatedly with anhydrous tetrahydrofuran (THF) and then dried at 70 °C under vacuum in order to remove any traces of adsorbed TEA on the surface of acylated MWCNT. This sample is labelled as MWCNT–COCl [25].

### 2.4. Modification of Multiwall Carbon Nanotube

Three separate solutions of hydroxylamine (0.2 g), cystine (0.5 g), and hydrazine (200 µL) were prepared in 1 mL THF and 0.5 mL pyridine. To each solution was added 0.05 g of MWCNT–COCl. The mixtures were stirred for 30 min at room temperature and then refluxed at 100 °C for 96 h. The residual hydroxylamine, cystine, and hydrazine were removed by washing with ethanol and sonication for 15 min. This washing process was repeated three times, until a clear ethanol was produced. The remaining solid was suspended in dichloromethane, sonicated and centrifugation. The collected black solid was dried under vacuum and labeled as (MWCNT–HA, MWCNT–CYS, MWCNT–HYD, as shown in Figure 1 [26].

### 2.5. Magnetization of Modified Multiwall Carbon Nanotube

The Magnetite/multiwall carbon nanotube functionalized with hydroxylamine, cystine, and hydrazine (m-MWCNT–HA, m-MWCNT–CYS, m-MWCNT–HYD) composite were synthesized by the combination of solution of 0.1 M ferric chloride hexahydrate (FeCl_3_·6H_2_O) and 0.05 M ferrous chloride tetrahydrate (FeCl_2_·4H_2_O), which was prepared with one to two molar ratios, then mixed with 0.05 g MWCNT and suspended for 2 h. After that, 5 M of NH_4_OH solution was used to precipitate iron oxides at 70 °C till pH was set at 10. The entire solution was kept under stirring for 1 h. The solid was allowed to cool down and magnet-separate, then washed with distilled water and ethanol. The compound collected was dried in a 2 h at 100 °C [27].

### 2.6. Instrument and Characterization

Instruments required for this research include: thermometer, shaking water bath (Daihan Labtech, Korea), 20 to 250 rpm digital speed control), pH scale (model: 3510, JENWAY, USA), flame atomic absorption spectroscopy (Thermo-scientific, iCE-3300,3000 series, USA), Fourier transform infrared spectroscopy (FTIR-SHIMADZU, Japan, Model: FTIR-8700). FT-Raman spectrometer (RFS 100/S – Bruker Inc., Karlsruhe, Germany) with a liquid-nitrogen-cooled germanium diode detector and an ND: YAG laser providing an impressive NIR line at 1064 nm was used for characterization. The spectra were recorded in a spectral area of 1 cm^−1^ to 4000 cm^−1^ at a spectral resolution of 2 cm^−1^ with 800 samples at approximately 30 mW laser power. Scanning Electron Microscopy (SU8000 Hitachi, Japan) at Peter Grunberg Institute for electronic materials (PGI-7), Julich, Germany, Transition Electron Microscopy, FEI Titan 80–300 TEM with a HRTEM at Ernst Ruska-Centrum (ERC), Julich, Germany. Brunauer–Emmett–Teller (Micromeritics, Norcross, GA, USA) for surface area, and thermogravimetric analysis (TGA Instruments, New Castle, DE, USA) from 20 to 700 °C, Vibrating Sample Magnetometer (VSM-LAKESHORE 7404, Boston, MA, USA). 

The chromatographic separation was done on an HPLC system, which is comprised of a mobile phase degasser unit (Model: DGU-20A3), 30 μL sample loops, pump (Model: LC-20AB), and a photo diode array detector (PDD) (Model: SPD-M20A), manufactured by Shimadzu Corporation, Japan. The mobile phase (60% acetonitrile and 40% of 0.2% formic acid in water) with flow rate of 0.8 mL min^−1^. The column used for separation was a C18 HPLC (250 mm × 4.00 mm × 5 µm). Limits of detection and quantification were 0.6 and 2.4, respectively. The wavelength of the detector was 220 nm for detecting ibuprofen [28]. Standard samples contained 5, 10, 15, 20, 30, and 50 ppm ibuprofen.

### 2.7. Batch Adsorption Studies

#### 2.7.1. Effect of Contact Time

Shaking time was analyzed to determine the optimal adsorption time. A 10 mL solution of Ibu with 20 ppm concentration was placed in a vial and shaken with 0.02 g of an adsorbent for various periods of time ranging from 1 to 120 min. The residual volume of the metal ion at the end of the time periods was measured using HPLC. 

#### 2.7.2. Effect of pH

The pH effect was tested using 0.1 M NaOH and 0.1 M HCl solutions to result in a pH ranging from 2 to 12. An amount of 0.02 g of adsorbent was applied in 10 mL of 20 ppm normal solution. After that, the final mixtures were put in a shaker inside water bath. The remaining amount of Ibu was determined after filtration using HPLC. 

#### 2.7.3. Effect of Temperature

To study the effect of temperature, 0.02 g of the adsorbent was added to 10 mL of the 20 ppm standard solution of Ibu at the optimum contact time and pH.

The range of temperature was from 5 to 50 °C. At the end of this time, the remaining amount of the Ibu was measured using HPLC.

#### 2.7.4. Effect of Metal Ion Concentration

Different standard concentrations of Ibu (10 mL) were used to find the optimum concentration. At the end of the time, the remaining amount of Ibu was measured using HPLC.

The amount of adsorbed of ibuprofen (*q_e_*, mg/g) was calculated from the equation [29]:(1)$$q_{e}=\frac{V\left(Co-Ce\right)}{W}$$
where *V* is the volume of the solution (L), $C_{o}$ is the initial Ibu concentration (mg/g), $C_{e}$ is Ibu concentration at equilibrium (mg/g), and *W* is the adsorbent mass (g). 

#### 2.7.5. Adsorption Isotherms Models

Adsorption isotherms are important for explaining adsorbate interaction with the adsorbent. The adsorption is performed to determine different conditions for adsorption. Langmuir, Freundlich, Temkin, Sips, etc. are the most common ones. Isotherm methods are used to explain the results of analysis of adsorption. Such models find essentially comparable guidelines with small differences in their strategies [30].

The Langmuir isotherm was applied to the absorption process on homogeneous surfaces and expressed as [31]:(2)$$\frac{C_{e}}{q_{e}}=\frac{1}{q_{m}}C_{e}+\frac{1}{q_{m}K_{L}}$$
where $C_{e}$ is the equilibrium concentration of adsorbate (mg/L), $q_{e}$ is the amount of adsorbate adsorbed per unit weight of adsorbent (mg/g), $q_{m}$ is the adsorption capacity (mg/g) or monolayer capacity, and $K_{L}$ is a constant (L/mg).

The Langmuir isotherm can be expressed by a constant dimensions separation factor ($R_{L}$), as shown by the following equation [32,33]:(3)$$R_{L}=\frac{1}{\left(1+K_{L}C_{o}\right)}$$
where $C_{o}$ is the highest initial concentration of adsorbate (mg/L), and $K_{L}$ (L/mg) is the Langmuir constant. 

The value of the $R_{L}$ refers to the form of the isotherm as being either unfavorable ($R_{L}$ > 1), linear ($R_{L}$ = 1), favorable (0 < $R_{L}$ <1), or irreversible if ($R_{L}$ = 0).

The Freundlich isotherm describes adsorption on a heterogeneous surface. The rate of adsorption/absorption varies with the strength of the energy at the adsorptive sites. Freundlich’s equation is expressed as follows:(4)$$\ln q_{e}=\frac{1}{n}ln\;C_{e}+lnK_{F}$$
where $K_{F\;}$ (mg/g) and 1/n are the constant characteristics of the system [34].

$K_{F\;}$ is an indicator of adsorption capacity of the adsorbent and 1/*n* being an indicator of favorability of the adsorption process. If (10 > *n* > 0), this denotes a favorable adsorption process.

Langmuir and Freundlich’s isotherms were used to define the relationship between the amounts of Ibu adsorbed on m-MWCNT–HA, m-MWCNT–CYS, m-MWCNT–HYD adsorbents and its equilibrium concentration in solutions.

#### 2.7.6. Adsorption Kinetics

Many kinetic adsorption models have been established to define the kinetic and speed levels. These models provide clues to the performance of the adsorption system and the removal rate of certain components using a particular adsorbent. It is also determined whether the adsorption process is physical or chemical in nature, and which step is the rate determining step. Examples of kinetic adsorption models include first-order pseudo models, second-order pseudo models, kinetic intraparticle diffusion model, reversible first order reaction model, Elovich’s model, etc. [35]

Pseudo-first-order kinetics were developed for describing adsorption kinetics, and are considered to be the earliest model. The equation for this model can be summarized as follows
(5)$$\log\left(q_{e}-q_{t}\right)=\log q_{e}-\left(\frac{K_{1}}{2.303}\right)t$$
where $q_{e}$ and $q_{t}$ are the masses of the adsorbate in equilibrium or at time *t* per unit mass of adsorbent (mg/g). *K*_1_ is the rate constant of the first-order pseudo-adsorption model (1/min).

The graph of log ($q_{e}-q_{t}$) as a function of t gives a straight line for first-order pseudo-adsorption with log *q**_e_* as the y intercept and (−$K_{1}$/2.303) as a slope [36].

The pseudo-second-order models are based on the idea that chemical adsorption could be the step to specify the rate, requiring the exchange or exchanging of electrons between adsorbent and adsorbent. The equation can be summarized as follows
(6)$$\frac{t}{q_{t}}=\frac{1}{q_{e}}t+\frac{1}{K_{2}q_{e}^{2}}$$
where $K_{2}$ is the equilibrium rate constant of the adsorption pseudo-second-order (g/mg∙min). The graph of *t*/$q_{t}$ versus t should give a linear relationship that allows the calculation of a second-order rate constant, $K_{2}$, of the intersection *Y* and $q_{e}$ of the slope [30,37,38].

The metal ion adsorption is mainly achieved through three steps, including migration of metal ion from liquid to surface of the adsorbents (film diffusion process), diffusion of metal ions within the porous structure (intraparticle diffusion process, and metal ion adsorption on the surface of the adsorbent. In general, the third step is quick and is not considered to be a rate-controlling step. 

An intra-particle diffusion kinetic model was suggested by Weber and Morris. The net equation of this kinetic model is:(7)$$q_{t}=K_{id}t^{0.5}+C$$
where $K_{id}$ is the constant diffusion rate (g/mg∙min^0.5^), and C is a constant representing the thickness of the boundary layer (mg/g). A graph of $q_{t}$ with respect to *t*^0.5^ will show a linear relationship with the constant C as the intersection in *y*. When the curve moves through the origin, adsorption is controlled by the process of interparticle diffusion.  Otherwise, it is dominated by film diffusion [39].

Activation energy can also be measured using the Arrhenius equation:(8)$$\mathrm{Ln}\;K_{2}=\mathrm{Ln}\;A-E_{a}/RT\;$$

In general, if $E_{a}$ is between 5 kJ/mol and 40 kJ/mol, the adsorption is physisorption, whereas a value of 40 kJ/mol to 800 kJ/mol indicates chemisorption.

#### 2.7.7. Adsorption Thermodynamics

The thermodynamic study was done by ascertaining enthalpy, free energy, entropy.

Thermodynamic parameters are required to conclude whether the process is spontaneous or not. Gibb’s free energy change, Δ*G*^0^, is an indication of the spontaneity of a chemical reaction and therefore is an important criterion for spontaneity. Both enthalpy (Δ*H*^0^) and entropy (Δ*S*^0^) factors must be considered to determine Gibb’s free energy of the process. Reactions occur automatically at a certain temperature if Δ*G*^0^ is a negative quantity. 

The following equation is the general equation that connect between thermodynamics parameters
(9)$$\Delta G^{0}=\Delta H^{0}-T\Delta S^{0}$$where *T* is the temperature in Kelvin (K).

The change in Gibbs energy can be expressed by the following equation: (10)$$\Delta G^{0}=–\;RT\;\ln\;K_{D}$$
where $K_{D}\;$ is the constant of thermodynamic equilibrium equal to ($q_{e}$/$c_{e}$) with a unit of (L/g). *R* is the gas constant, 8.314 J/mol∙K. 

The net equation of the last two equations can be expressed as the following equation:(11)$$lnK_{d}=-\frac{\Delta H^{0}}{RT}+\frac{\Delta S^{0}\;}{R}$$

The plot of In *K_d_* against (1/*T*) gives a line with ($\Delta S^{0}\;$/*R*) as y-intercept and (−$\Delta H^{0}$/*R*) like the slope. The resulting graph is identified as a Van’t Hoff diagram [40,41,42]. 

#### 2.7.8. Regeneration Process

The adsorbent after the adsorption process is washed with 0.1 M NaOH solution, then with distilled water, and then let dry for 24 h. The same technique of recovery is then used for each regenerated adsorbent in order to prove that the three modified MWCNT can be used for several times with no effect on the efficiency of removal of Ibu.

## 3. Results and Discussion

### 3.1. Material Characterization

#### 3.1.1. SEM Analysis of the Modified m-MWCNT

Figure 2 shows the MWCNT before and after modifications. The SEM images indicate that all samples exhibit forest-like morphologies, and it is clear that the unmodified tubes and the modified ones are randomly oriented. Additionally, it is obvious from the SEM images and from the diameter distribution that the diameter of the modified carbon nanotubes is slightly greater than that of the original ones, with the largest tube diameter being obtained for MWCNT–COOH (Figure 2b). Moreover, the length of the modified MWCNT was reduced compared to the unmodified MWCNT. In the case of the unmodified MWCNT (Figure 2a), it is obvious that the tubes are long, thin and did not agglomerate together a lot, while in the case of the modified MWCNT, the length of the tube decreased and it became thicker and tended to agglomerate together. The diameters of the obtained nanotubes ranged from 0.062 to 0.146 μm. The average diameter of the carbon nanotubes depends on the modifying group. The smallest average diameter of MWCNT (0.062 ± 0.92) was obtained for m-MWCNT–HYD. The quite large standard deviation of the average diameter of the nanotubes is a result of the tendency of the powder to agglomerate.

#### 3.1.2. Transmission Electron Micrograph (TEM) Analysis

Figure 3a shows the TEM results of the unmodified carbon nanotubes used in the modification process. The nanotubes used are homogenous. The TEM images of the modified nanotubes consisting of carbon nanotubes decorated with novel functionality are presented in Figure 3b–e. Based on these images, it can be seen that the functional groups used to decorate the nanotube in the decoration process are arranged on the surface of the carbon nanotubes. The increase in the modified nanotube diameter is clearly observable in the TEM images. It is important to note that the tube diameters as obtained from TEM images are a close match for the SEM results. 

#### 3.1.3. BET Analysis

The measurement of the surface area of the carbon nanotubes is dependent on the absorption of N_2_ gas. The BET model (Brunauer–Emmett–Teller) was used. This was applied to isothermal absorption N_2_ at 77 K (liquid temperature N_2_). The Langmuir adsorption theory can be extended to describe the multi-layer adsorption of N_2_ onto the surface of the material. BET theory equals the rate of monolayer condensation from condensation to the previous monolayer absorption rate on the surface.

The activation or chemical treatment of MWCNT used for purification and treatment can open the covered ends of the nanotubes. This allows the molecules to be absorbed from the nanotubes, thus making it possible to measure them. MWCNT surface area is no longer the surface area of the exterior and may also include surface area within the nanotubes. The ends of the tube were removed using different chemical and thermal stimulation methods, so the effective surface area increased as shown in Table 1.

#### 3.1.4. FT-IR Characterization

An FT-IR analysis also confirmed the modification of MWCNT, as shown in Figure 4. Comparison of MWCNT FT-IR spectra and oxidized MWCNT revealed the presence of broad bands at 3500 cm^−1^, corresponding to OH attached matrix, and 1700 cm^−1^, corresponding to C=O stretching carboxylic acid group vibration, which suggested that the carboxylic group on MWCNT had been successfully prepared. When HA, CYS, and HYD were added, some new peaks appeared in the spectrum. The new peaks can be attributed as follows, on the basis of the literature: the peak at 1640 cm^−1^ is due to C=O stretching vibration. C–N stretching vibration and N–H bending vibration appear at 1550 cm^−1^. 

#### 3.1.5. Raman Characterization

The Raman spectrum of MWCNT shows two prominent bands at 1604 cm^−1^ (G band), which is assigned to the in-plane vibration of the C–C bond, and at 1291 cm^−1^ (D band), which is related to a disorder in carbon systems (Figure 5). The D band is a characteristic band of carbon nanostructures. Comparing the spectra of the oxidized MWCNT to that for MWCNT, it can be clearly noticed that the D-band of m-MWCNT–COOH shifted from 1360 to 1349 cm^−1^, and G-band from 1605 to 1600 cm^−1^, and this shows that a large defection was created after oxidation. Additionally, the ratio of the intensity of the D-band peak to that of the G-band peak (ID/IG) of the pure MWCNT was 1.33, while that for the MWCNT–functionalized with HA, CYS, HYD was 1.71. This is an indication that HA, CYS, and HYD had been successfully functionalized onto the MWCNT surface.

#### 3.1.6. TGA Analysis and Thermal Stability

The analysis was performed under air with a heating rate of 10 °C/min during the entire analysis. The results of the TGA analysis are shown in Figure 6. The TGA results reveal that the raw MWCNTs are stable and did not undergo any weight loss at temperatures below 600 °C. Meanwhile, the modified MWCNTs exhibited some weight loss at about 170 °C. This is another indication of the presence of thermally unstable functional groups.

#### 3.1.7. VSM Characterization

The magnetic properties of the prepared carbon nanotubes (m-MWCNT–HA, m-MWCNT–CYS, and m-MWCNT–HYD) were measured using a vibrating-sample magnetometer VSM with a maximum field of 9 T. Figure 7a shows the resulting hysteresis curves for all measured samples. It is obvious that the carbon nanotubes modified with hydrazine (m-MWCNT–HYD) show a higher magnetization value of 1.4 emu/g, while the magnetization values for the carbon nanotubes modified with hydroxyl amine (m-MWCNT-HA), and 2-amino-3-mercaptopropanoic acid (m-MWCNT–CYS) are 0.84 emu/g, and 0.44 emu/g, respectively. The maximum magnetization, Mm, equals the saturation magnetization, Ms. For the carbon nanotubes without any modification (MWCNT), the magnetization curve has a semi-linear appearance, which indicates a paramagnetic state. For m-MWCNT–HA, m-MWCNT–CYS and m-MWCNT–HYD, the samples show a sigmoidal response, but show no hysteresis, which is an indication of the presence of a saturated superparamagnetic component in the samples. Furthermore, m-MWCNT–HA displayed a very weak ferromagnetic behavior, as shown in Figure 7b, with a small coercive field of approximately 7 Oe. The preceding results indicate an important phase transition from paramagnetic to a weak ferromagnetic state with novel functionality of the multiwall carbon nanotube. 

### 3.2. Investigation of Ibuprofen Adsorption Parameters

Adsorption of ibuprofen using different adsorbents like (m-MWCNT–HA), (m-MWCNT–CTS) and (m-MWCNT–HYD) were studied. To determine the maximum adsorption efficiency, several parameters were studied, including contact time, pH, temperature, etc. 

#### 3.2.1. Effect of Contact Time on Ibu adsorption

For the influence of contact time, values ranging from 5 to 120 min were studied. Other parameters used were an initial concentration of 20 ppm, 0.02 g adsorbents and 25 °C temperature. The results of this process are shown in Figure 8a. From this figure, it can be seen that the highest percentage of Ibu removal was 56.5% using m-MWCNT–HA and m-MWCNT–HYD after 30 and 15 min, while when using m-MWCNT–CYS, the removal was 55% after 15 min.

Therefore, 15- and 30-min contact times were chosen for the rest of the trials.

#### 3.2.2. Effect of pH on Ibuprofen Adsorption

The adsorption efficiency of m-MWCNT–HA, m-MWCNT–CYS, m-MWCNT–HYD as a function of pH was also evaluated. The pH value ranged from 2.0 to 11.

The maximum adsorption efficiencies on m-MWCNT–HA, m-MWCNT–CYS, m-MWCNT–HYD were found at pH = 4, and the percentages of removal were 61.5%, 84.5%, 98.4% for m-MWCNT–HA, m-MWCNT–HYD, m-MWCNT–CYS, respectively.

The pH effect could be explained in terms of zero-point charge of adsorbents. Mass titration technique (MT) was used to assess the pHpzc values of the adsorbents [43]. The pHpzc results are shown in Figure 8c, and the obtained zero-point charges are 3.5, 3.7, 3.2 for m-MWCNT–HA, m-MWCNT–CYS, m-MWCNT–HYD, respectively, with maximum adsorption occurring at pH > pHpzc. At pH higher than pHpzc, Ibu is deprotonated, as shown in Figure 9, and a negative charge develops on the carboxyl group. The presence of negatives leads to strong dipole–dipole interaction (H-bonding) between the adsorbent functional group (N–H) and the adsorbate anion. A schematic representation of this interaction is shown in Figure 9. In addition to the previously mentioned attraction force, a secondary attraction force could also contribute to adsorption efficiency, as represented by π–π interactions between the pi-electrons of the Ibu phenyl ring and the pi-electrons of MWCNT, which is reported in the literature as a main factor in certain adsorption processes [44,45]. Additionally, n–π interactions could be considered a contribution factor, especially those occurring between the lone pair of electrons of oxygen and nitrogen (–OH, –NH) and the unhybridized p-orbitals of aromatic π bonds [46,47,48]. Another crucial factor that should also be considered in the adsorption process in this case is the insertion of Ibu in the MWCNT pores and getting trapped inside. This type of adsorption is considered pH-independent, but it might be enhanced at low pH, were the Ibu is in a protonated state. 

At pH = 4, both adsorbent and adsorbate behave as H-bonding donor and acceptor, and this leads to higher H-bonding; at higher pH, the adsorbate behaves as an H-bonding acceptor only, and this leads to a weaker interaction. Furthermore, at higher pH, the electron densities of both are high, and this can cause a kind of repulsion. 

The same type of interaction appears when using m-MWCNT–HA and m-MWCNT–CYS.

#### 3.2.3. Effect of Temperature on Ibu Adsorption

Temperatures ranging from 5 to 50 °C were used. Figure 8d shows that the highest percentages of removal for the three adsorbents occurred at a temperature of 25 °C, and the adsorption started to decrease when increasing the temperature beyond 25 °C.

#### 3.2.4. Effect of Ibu Initial Concentration

Figure 8e shows that the percentage of Ibu removal decreases with the increase in initial Ibu concentration. The highest percentage of Ibu removal was 98.4% for m-MWCNT–CYS, while m-MWCNT–HYD and m-MWCNT–HA exhibited percentage removals of 93% and 61.5%, respectively. In general, low initial concentrations of specific components result in enough adsorption sites to be available.

### 3.3. Equilibrium Isotherm Models for Ibu Adsorption

The Langmuir and Freundlich isotherms were used to define the relationship between the amounts of Ibu adsorbed on m-MWCNT–HA, m-MWCNT–CYS, and m-MWCNT–HYD adsorbents and their equilibrium concentration in the solutions.

Figure 10a shows the Langmuir isothermal adsorption parameters examined by plotting $C_{e}$/$q_{e}$ against $C_{e}$, while (Figure 10b) shows the Freundlich adsorption isotherm parameters examined by plotting ln$q_{e}$. against ln$C_{e}$.

Langmuir’s and Freundlich’s model parameters are summarized in Table 2. These values confirmed the applicability of the Langmuir model preference. The values of Langmuir constant $q_{m}$ (mg/g) were 1.15, 7.56 and 11.8 for m-MWCNT-HA, m-MWCNT-CYS, and m-MWCNT-HYD, respectively. The values of constant $K_{L}$ (L/mg) were 39.5, 0.567 and 0.26. The dimensionless constants were 0.001, 0.08 and 0.16 for m-MWCNT-HA, m-MWCNT-CYS, and m-MWCNT-HYD, respectively; R^2^ (0.99, 0.94, 0.87). All these data indicate good sorption.

For the Freundlich constants, *n* and $K_{F\;}$ (mg/g) are Freundlich coefficients related to the relative adsorption capability and sorption rate. The *R*^2^ values (0.87, 0.03, 0.39) have also been verified as not being favorable. 

As shown from the previous figures, the values of *R*^2^ are approximately 1 when using the Langmuir isotherm and the value of $R_{L}$ is between zero and 1. This means that the adsorption of Ibu at (m-MWCNT–HA, m-MWCNT–CYS, m-MWCNT–HYD) can be fitted using the Langmuir isotherm.

### 3.4. Adsorption Kinetic Model

Kinetic parameters for the pseudo-first-order, pseudo-second-order and intra-particle diffusion were determined to investigate the mechanism of the adsorption process. 

The kinetic adsorption model is shown in Figure 11a–d. The pseudo-first-order rate constant and regression coefficient (R^2^) were 0.34, 0.68, 0.006 for m-MWCNT–HA, m-MWCNT–CYS, m-MWCNT–HYD, respectively. The experimental and theoretical values of equilibrium amounts were 5.5, 5.61, 45.43 and 0.71, 2.54, 0.3 mg/g, with high differences indicating the non-applicability of the pseudo-first-order reaction.

However, the pseudo-second-order kinetic model was also applied to the experimental data, while R^2^ in pseudo-second-order is approximately one.

The experimental and theoretical values of equilibrium amounts were 5.5, 5.61, 45.43 and 5.8, 6.02, 5.65 mg/g, with very low differences, i.e., they were close to each other.

By comparing the $q_{e}$. (experimental) and the $q_{e}$ (calculated) values, we inferred that the experimental values for all adsorptions were close to the *q_e_* (calculated) values in the second-order adsorption model. In addition, this model reflects the mechanism for adsorption. Kinetic parameters summarized in Table 3.

One important test for determining the type of adsorption is to measure activation energy. From the value of $E_{a}$, the adsorption was physical in nature (Figure 11d).

### 3.5. Adsorption Thermodynamics

The Van’t Hoff plot was used to calculate the common thermodynamics parameters: $\Delta S^{0}$, $\Delta H^{0}$, and $\Delta G^{0}$ for Ibu adsorption on m-MWCNT–HA, m-MWCNT–CYS, m-MWCNT–HYD, as shown in Figure 12.

As shown in Table 4, the values of free energy were negative. This sign is a clear indication of spontaneous and favorable adsorption, while the values of enthalpy were positive. This is a clear indication of the endothermic process.

The positive $\Delta S^{o}\;$ values reveal an increase in randomness at the solid and liquid surface, indicating the accumulation of Ibu.

### 3.6. Adsorbent Regeneration

Economic adsorption status was confirmed by the recycling method. For this, adsorbent regeneration was achieved by using dilute sodium hydroxide. Figure 13 shows the effect of absorbed recovery on the absorption of Ibu on m-MWCNT–HA, m-MWCNT–CYS and m-MWCNT–HYD.

As shown in Figure 13, the difference between the percentages of Ibu removal after the second and third regeneration is very low. This is strong evidence that the three compound absorbents can be recycled, and therefore can be used multiple times.

## 4. Conclusions

In this study, a magnetic multiwall carbon nanotube functionalized with hydroxylamine, cystine and hydrazine was synthesized and tested for ibuprofen removal from water. The efficiency of the prepared derivatives toward Ibu was studied as a function of pH, metal ion initial concertation, temperature and time. The three adsorbents showed excellent efficiency toward Ibu and % of removal was quantitative. The highest efficiency was determined to be at room temperature and a pH of 4.0. The kinetic study revealed that the Ibu adsorption by the three adsorbents followed pseudo-second-order and Langmuir isotherm models. The thermodynamic analysis showed a negative free energy, indicating a spontaneous adsorption process at room temperature. The adsorbents were regenerated by treatment with 0.1M NaOH. The efficiency of the regenerated adsorbents showed no change even after three cycles of regeneration and reuse.

## Figures and Tables

**Figure 1 materials-13-03329-f001:**
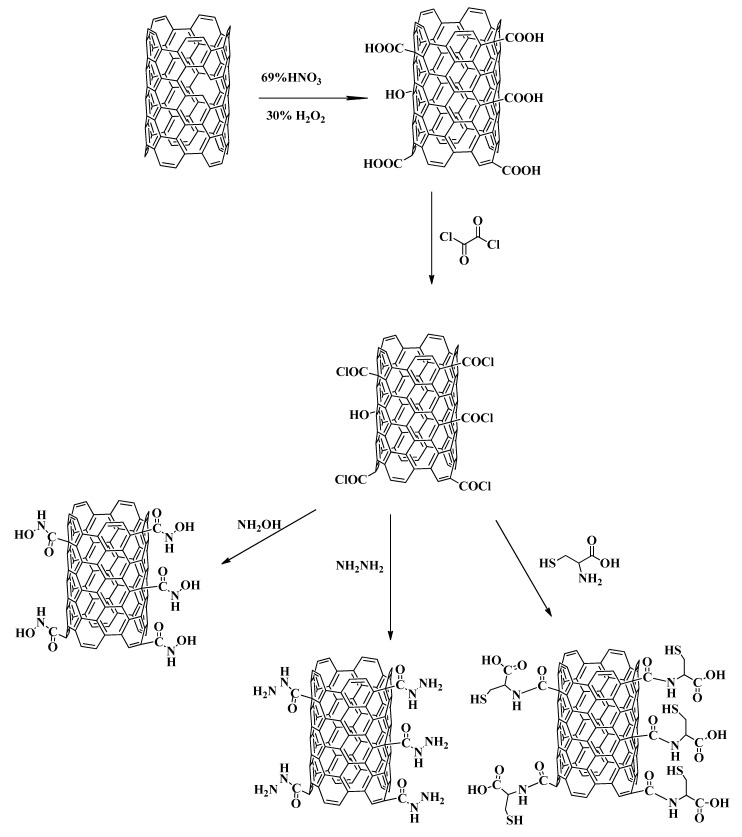
Preparation of magnetic functionalized multiwall carbon nanotube.

**Figure 2 materials-13-03329-f002:**
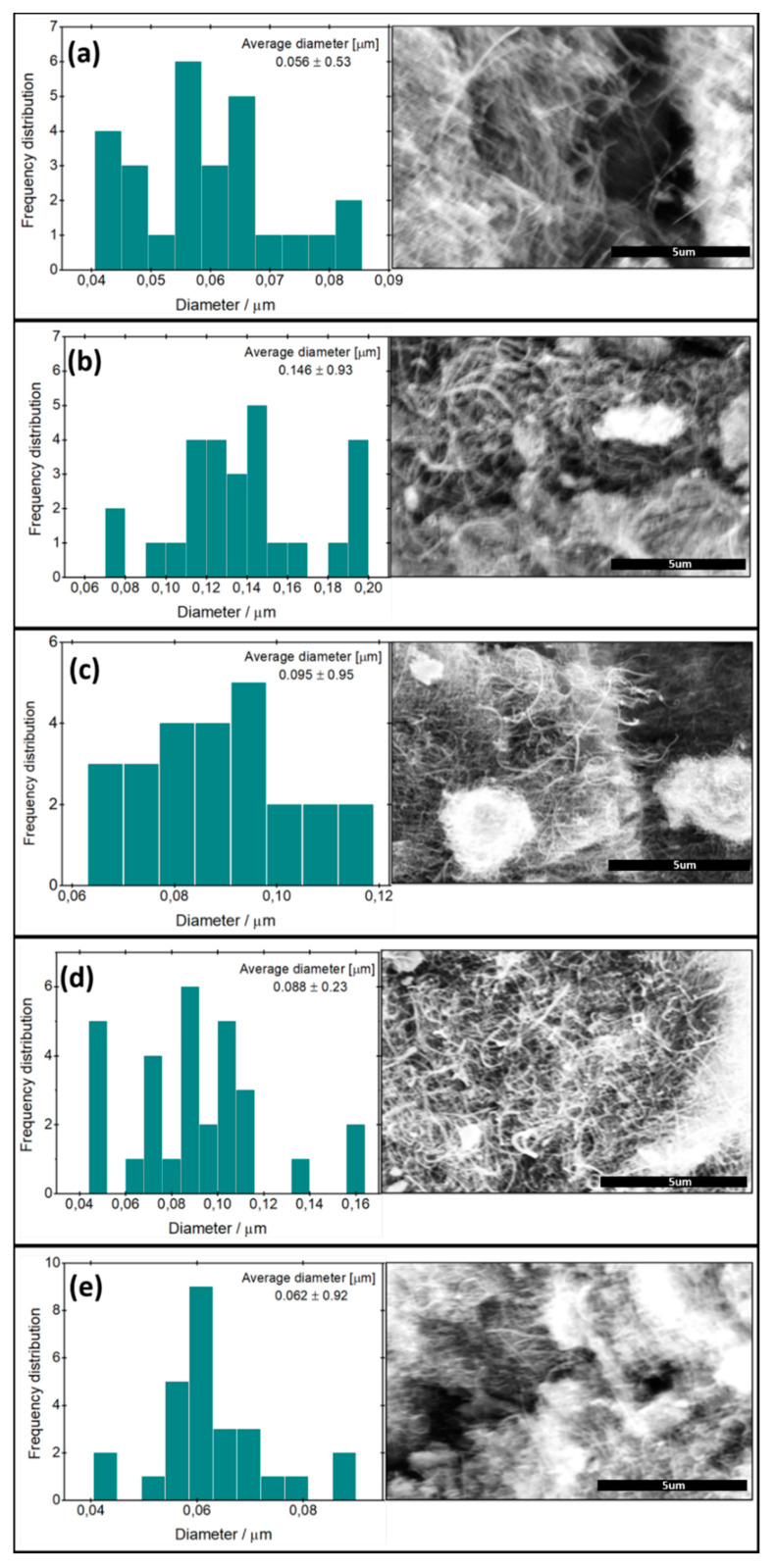
SEM images of the original and the modified carbon nanotube (right), together with diameter distribution (left). (**a**) MWCNT, (**b**) MWCNT–COOH, (**c**): m-MWCNT–HA, (**d**) m-MWCNT–CYS, and (**e**) m-MWCNT–HYD. Scale bars: 5 μm.

**Figure 3 materials-13-03329-f003:**
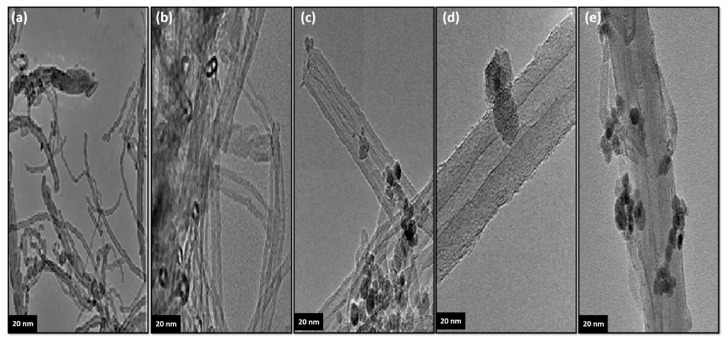
TEM images of the unmodified and the modified carbon nanotubes: (**a**) MWCNT, (**b**) MWCNT–COOH, (**c**) m-MWCNT–HA, (**d**) m-MWCNT–CYS and (**e**) m-MWCNT–HYD. Scale bars: 20 nm.

**Figure 4 materials-13-03329-f004:**
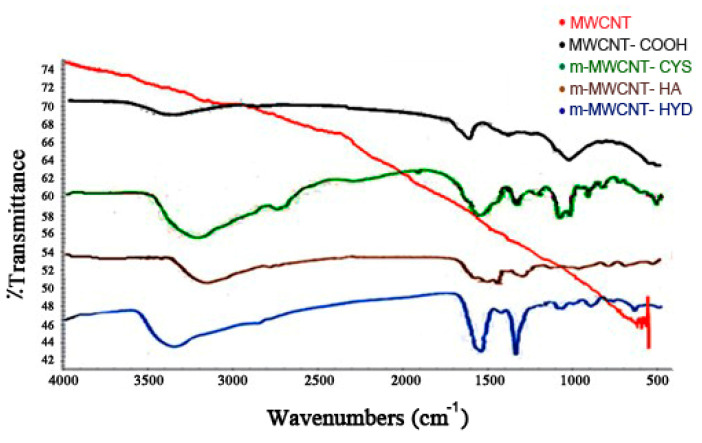
FT-IR spectra for MWCNT, MWCNT–COOH, m-MWCNT–HA, m-MWCNT–CYS, and m-MWCNT–HYD.

**Figure 5 materials-13-03329-f005:**
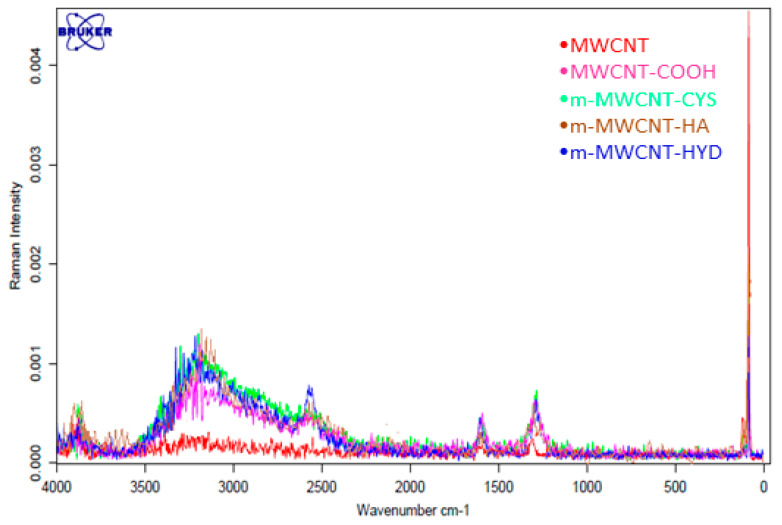
Raman spectra for MWCNT, MWCNT–COOH, m-MWCNT–HA, m-MWCNT–CYS, and m-MWCNT–HYD.

**Figure 6 materials-13-03329-f006:**
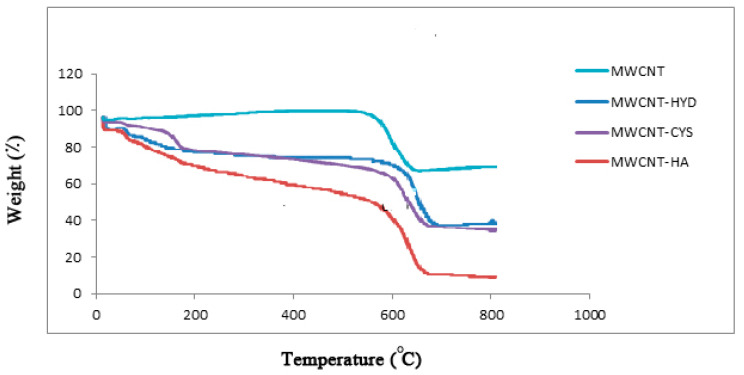
TGA analysis for MWCNT, m-MWCNT–HA, m-MWCNT–CYS, and m-MWCNT–HYD.

**Figure 7 materials-13-03329-f007:**
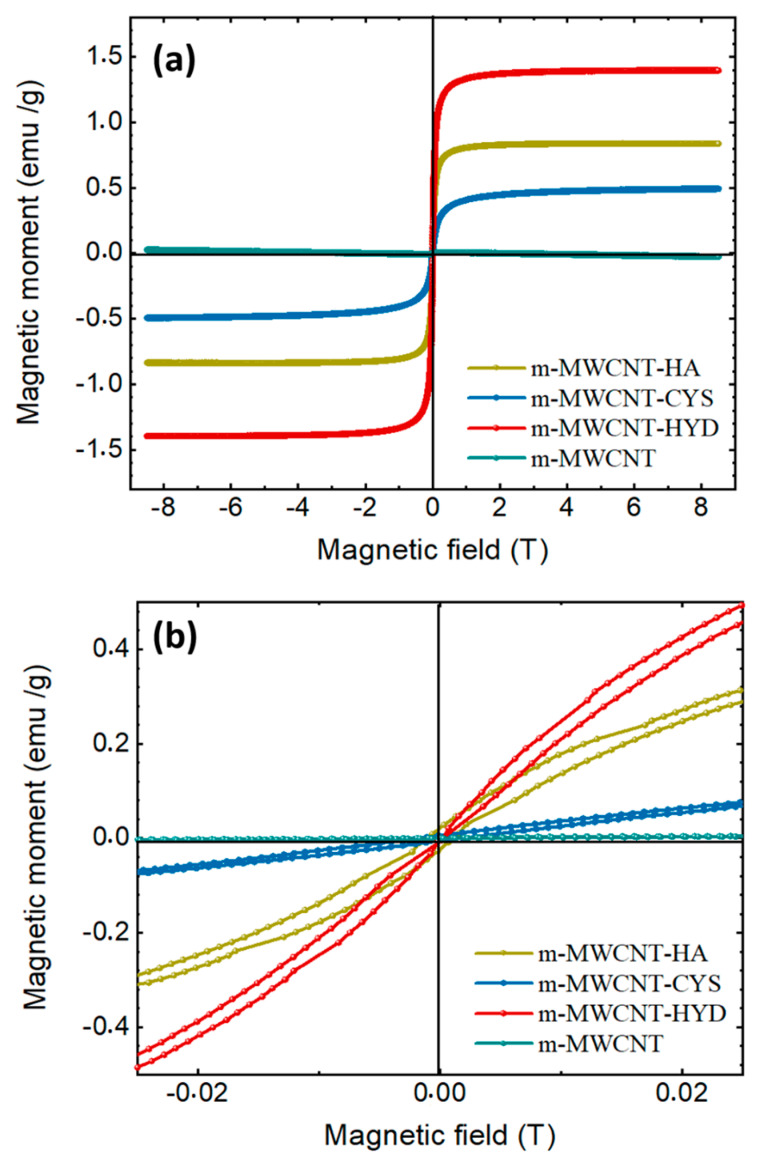
Magnetic field dependence of the magnetization measured at (**a**) 300 K (M–H loops) of m-MWCNT–HA, m-MWCNT–CYS, m-MWCNT–HYD and MWCNT; (**b**) the magnification of the central area of the hysteresis loops.

**Figure 8 materials-13-03329-f008:**
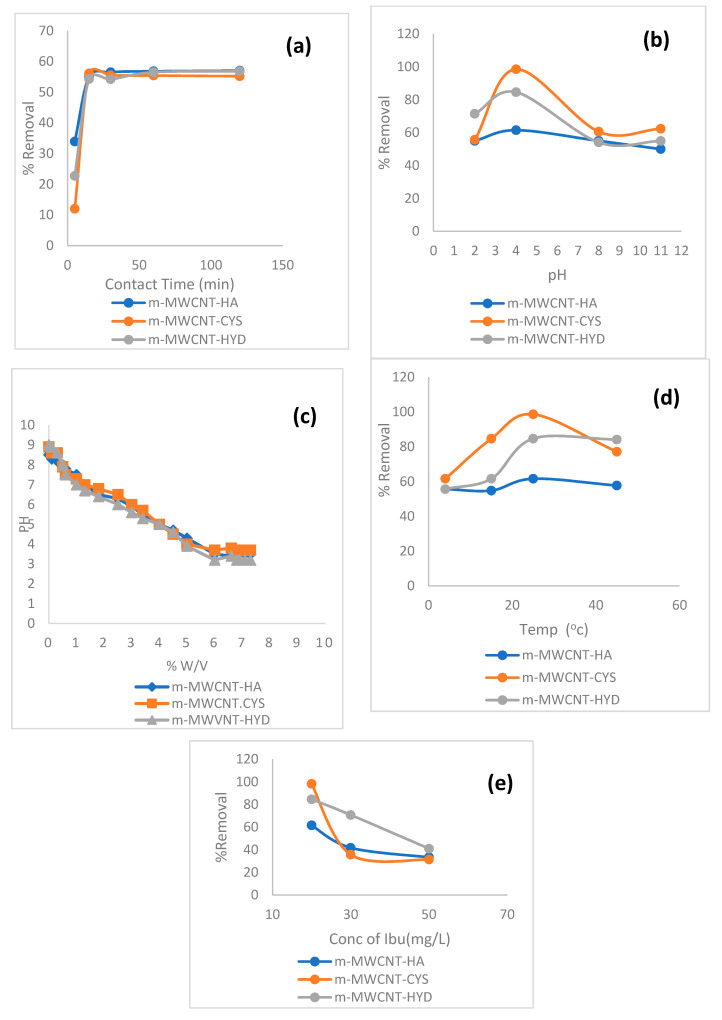
Effect of (**a**) contact time, (**b**) pH, (**c**) experimental mass titration curves, (**d**) temperature, and (**e**) Ibu initial concentration on Ibu adsorption on m-MWCNT–HA, m-MWCNT–CYS, and m-MWCNT–HYD.

**Figure 9 materials-13-03329-f009:**
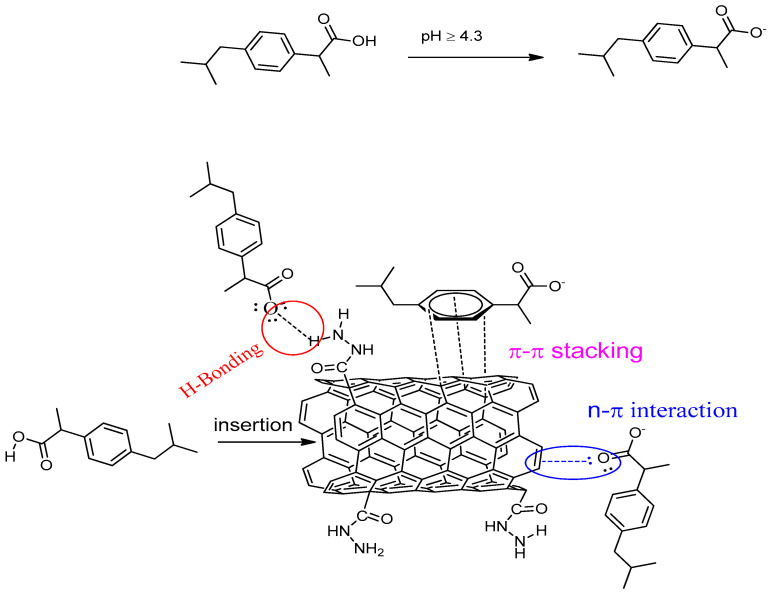
Mechanism of Ibu adsorption on m-MWCNT–HYD.

**Figure 10 materials-13-03329-f010:**
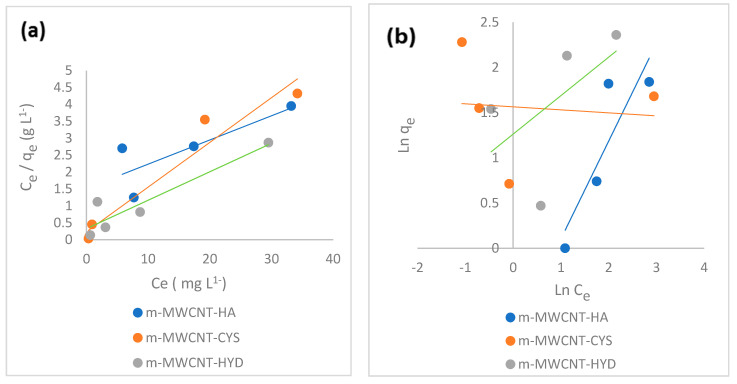
(**a**) Langmuir isotherm plot and (**b**) Freundlich isotherm plot for Ibu adsorption on m-MWCNT–HA, m-MWCNT–CYS, and m-MWCNT–HYD.

**Figure 11 materials-13-03329-f011:**
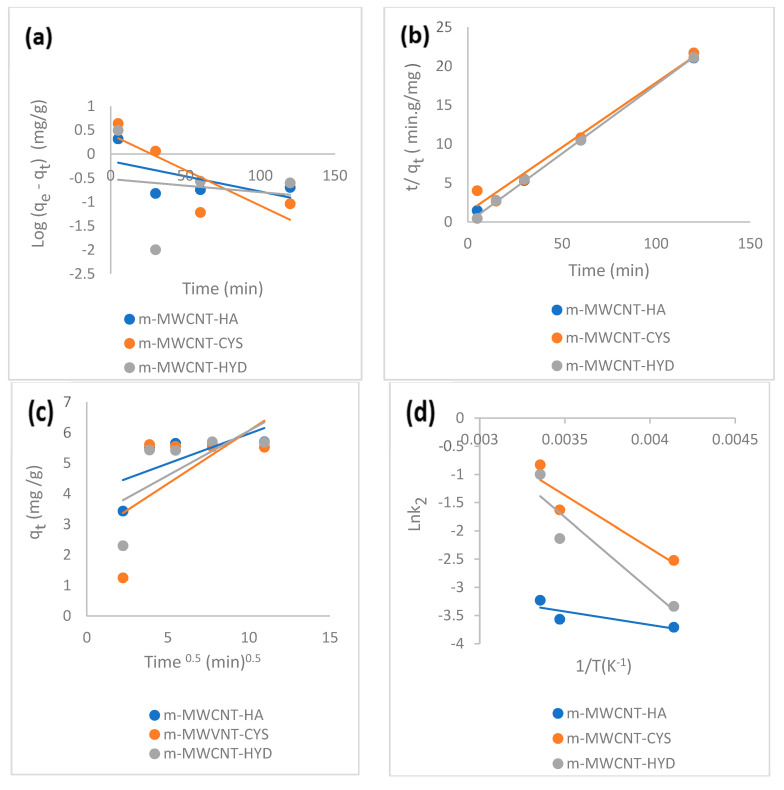
(**a**) The pseudo-first-order kinetic plot. (**b**) The pseudo-second-order kinetic plot. (**c**) Intra-particle diffusion kinetic model. (**d**) Arrhenius plot for Ibu adsorption on m-MWCNT–HA, m-MWCNT–CYS, and m-MWCNT–HYD.

**Figure 12 materials-13-03329-f012:**
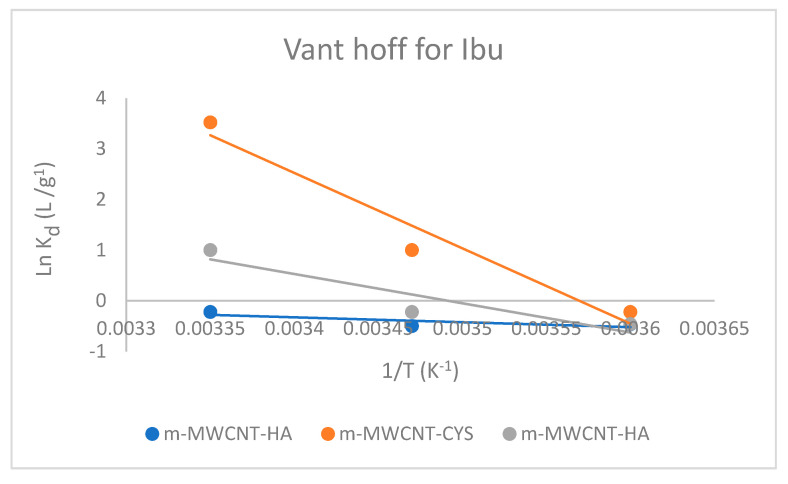
The Van’t Hoff plot for Ibu adsorption on m-MWCNT–HA, m-MWCNT–CYS, m-MWCNT–HYD.

**Figure 13 materials-13-03329-f013:**
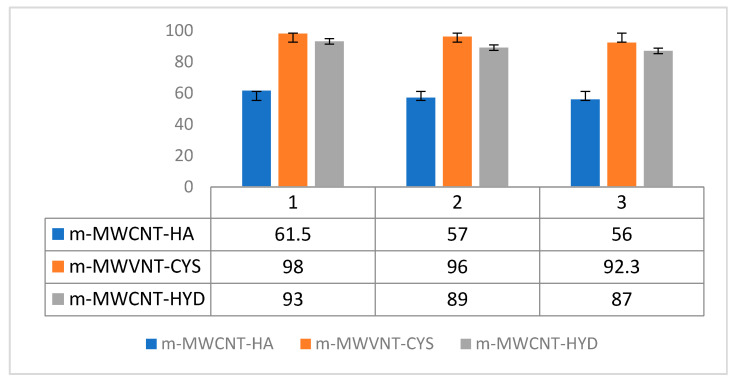
Three trials of adsorption–desorption of MWCNT using different functional groups towards removal efficiency of Ibu.

**Table 1 materials-13-03329-t001:** BET results for MWCNT, MWCNT–COOH, m-MWCNT–HA, m-MWCNT–CYA, and m-MWCNT–HYD.

SAMPLe	Multipoint Bet Area (m²/g)
MWCNT	111.5
MWCNT–COOH	175.1
m-MWCNT–HA	151.0
m-MWCNT–CYA	154.5
m-MWCNT–HYD	187.0

**Table 2 materials-13-03329-t002:** The Langmuir and Freundlich parameters for Ibu adsorption on m-MWCNT–HA, m-MWCNT–CYS, and m-MWCNT–HYD.

Adsorbents	Langmuir Isotherm				Freundlich Isotherm		
	*q_m_* (mg/g)	*K_L_* (L/mg)	*R_L_*	*R* ^2^	*K_F_* (mg/g)	*n* (g/L)	*R* ^2^
m-MWCNT–HA	1.15	39.5	0.0012	0.99	0.56	1.18	0.87
m-MWCNT–CYS	7.56	0.567	0.08	0.94	4.95	19.2	0.03
m-MWCNT–HYD	11.8	0.26	0.16	0.87	3.66	16.26	0.39

**Table 3 materials-13-03329-t003:** Parameters for Ibu adsorption of kinetic models of pseudo-first-order, pseudo-second-order and intraparticle diffusion on m-MWCNT–HA, m-MWCNT–CYS and m-MWCNT–HYD.

Adsorbents	Pseudo-First-Order Kinetics				Pseudo-Second-Order				Intra-Particle Diffusion		
	*q_e_* exp	*q_e_* (mg/g) Calc	*K*_1_ (1/min)	*R* ^2^	*q_e_* (mg/g)	*K*_2_ (g/mg.min)	*E_a_* (KJ)	*R* ^2^	C (mg/g)	*K_id_* (g/mg∙min^0.5^)	*R* ^2^
m-MWCNT–HA	5.5	0.71	0.014	0.34	5.8	0.11	4	0.99	4.007	4.007	0.45
m-MWCNT–CYS	5.61	2.54	0.03	0.68	6.02	0.021	15.5	0.97	2.59	2.59	0.37
m-MWCNT–HYD	5.43	0.3	0.006	0.017	5.65	0.368	21.3	0.99	3.13	3.13	0.46

**Table 4 materials-13-03329-t004:** Thermodynamic parameters for the adsorption of Ibu on m-MWCNT–HA, m-MWCNT–CYS, and m-MWCNT–HYD.

Adsorbents	$\Delta H^{0}\;(\mathrm{kJ})$	$\Delta S^{0}\;\;(J/K)$	$\Delta G^{0}\;(25\;{}^{\circ}C)$
m-MWCNT–HA	2.45	12.4	−1.20
m-MWCNT–CYS	123.73	441.6	−7.86
m-MWCNT–HYD	48.09	167.9	−1.94

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
