# Peer review of "Enhanced Ibuprofen Adsorption and Desorption on Synthesized Functionalized Magnetic Multiwall Carbon Nanotubes from Aqueous Solution"

_materials, 2020, doi:10.3390/ma13153329_

Round 1

Reviewer 1 Report

The existence of ibuprofen in water can have traumatic effects on living organisms which is reported to stop the cell reproduction in human embryos and has negative effect on aquatic vertebrate reproduction. CNTs are widely studied for water treatment and heavy metal adsorption. In this manuscript, the authors prepared a kind of magnetic multiwall carbon nanotubes to remove the ibuprofen from aqueous solution. The authors claim that the CNT synthesized in this work is featuring with magnetic recycling which may avoid nano-adsorbents from discharging into the natural environment and minimize future hazards. However, scientifically, this work do not show new mechanism on contaminant removal perspective. It seems that the only novelty comes from the magnetic function. Due to the unclear scientific significance of this work, the reviewer would like to recommend the authors to re-organized this manuscript and re-submit it to materials after addressing the important comments.

Comments:

  1. The introduction part should be re-orgnized and re-write because the current version does not show the novelty of this work clearly. Especially, from line 83 to line 102, the authors summarized recent progress on the removal of hazards inside water by CNT materials. However, the authors do not demonstrate the breakthrough of this work.
  2. The SEM images in Figure 2 and Figure 3 are stretched, the width and length of those images should be kept as the original ratio. Please revise it.
  3. For CNT synthesized in this work, what is the longest cycles can be used? How about the removal efficiency of ibuprofen in each cycles. The authors can test more cycles and add the data into Figure 13.
  4. The diagrams are badly organized in this manuscript, for example, Figure 10a, the title of y axis is overlapped with the values on y axis, it is not easy to read. The fitting lines in Figure 10, Figure 11, Figure 12, are messed up, please explain a little bit on how the authors make those fitting lines?
  5. Please add error bars into Figure 13.

Author Response

Comments:

1. The introduction part should be re-orgnized and re-write because the current version does not show the novelty of this work clearly. Especially, from line 83 to line 102, the authors summarized recent progress on the removal of hazards inside water by CNT materials. However, the authors do not demonstrate the breakthrough of this work.  Done

2. The SEM images in Figure 2 and Figure 3 are stretched, the width and length of those images should be kept as the original ratio. Please revise it.  Done

3. For CNT synthesized in this work, what is the longest cycles can be used? How about the removal efficiency of ibuprofen in each cycles. The authors can test more cycles and add the data into Figure 13. Three trials of adsorption – desorption, the difference between the percentages of Ibu removal after the second and third regeneration is very low

For MWCNT-HA: the percentage of removal was 61.5% and became 57% and 56% after second and third cycle

 For MWCNT-CYS: the percentage of removal was 98% and became 96% and 92.3 after second and third cycle

For MWCNT-HYD: the percentage of removal was 93 and became 89% and 87% after second and third cycle

4. The diagrams are badly organized in this manuscript, for example, Figure 10a, the title of y axis is overlapped with the values on y axis, it is not easy to read. The fitting lines in Figure 10, Figure 11, Figure 12, are messed up, please explain a little bit on how the authors make those fitting lines?

The labels of Fig 10a has been modified. The fittings were based on the equations from excel were slopes and intercepts have been used.

5. Please add error bars into Figure 13. Done

Reviewer 2 Report

Dear author,

Here are my comments:

  1. 3.1.1. Equipment description should be shown in Materials and Methods section not in Results and discussions.
  2. 3.1.2. TEM description should be place in Materials and methods section.
  3. 3.1.5. RAMAN description should be also moved.
  4. 3.1.7. VSM description should be moved.
  5. I think you should explain in more details the novelty of the paper (it is stated something at the beginning).
  6. What are the possible costs for your systems in Ibu removal? Is this cost-effective?

Author Response

Comments and Suggestions for Authors

Dear author,

Here are my comments:

  1. 3.1.1. Equipment description should be shown in Materials and Methods section not in Results and discussions. Done (The descriptions of all instruments have been moved to materials and chemicals. The only thing left is the results)
  2. 3.1.2. TEM description should be place in Materials and methods section. Done
  3. 3.1.5. RAMAN description should be also moved. Done
  4. 3.1.7. VSM description should be moved. Done
  5. I think you should explain in more details the novelty of the paper (it is stated something at the beginning). Done
  6. What are the possible costs for your systems in Ibu removal? Is this cost-effective?

Regeneration availability is an important factor to reduce the cost. We can use the adsorbent several time without reduces the efficiency. The method is capable to work under normal condition using low adsorbent dose.  Also, it is expected to greatly reduce in the price of MWCNT due to increase in commercial production.

Possible cost is 10 USD for each 1000 L of wastewater including regenerations.

Round 2

Reviewer 1 Report

Thank Authors for addressing some of the comments from the reviewer.

However, there still be some inherent problems existing in this manuscript which are not addressed. Those issues are critical for readers to understand the field of this work. Therefore, the reviewer suggest the authors to address those comments seriously before its publication in Materials.

  1. The SEM images in Figure 2 are not clear to read, and the TEM images in Figure 3 are stretched. Please give the high resolution SEM images in Figure 2, and give the original TEM images in Figure 3 with original width and length ratio.
  2. The novelty of this work is still not clearly demonstrated in the introduction part even though the authors did some minor changes based on the original version. Please re-organize the introduction part to highlight the novelty of this work.

Author Response

Rev 1 However, there still be some inherent problems existing in this manuscript which are not addressed. Those issues are critical for readers to understand the field of this work. Therefore, the reviewer suggest the authors to address those comments seriously before its publication in Materials.

1. The SEM images in Figure 2 are not clear to read, and the TEM images in Figure 3 are stretched. Please give the high resolution SEM images in Figure 2, and give the original TEM images in Figure 3 with original width and length ratio.

We just uploaded the original SEMs and TEMs as obtained with better resolutions

2. The novelty of this work is still not clearly demonstrated in the introduction part even though the authors did some minor changes based on the original version. Please re-organize the introduction part to highlight the novelty of this work.

We added this paragraph

The novelty of this work can be summarized as to the best of our knowledge, the grafted MWCNT’s presented in this work is the first example in the literature of oxidized MWCNT modified with such functionalities and applied for Ibuprofen removal. Other fact that previous studies on removal of IBU using one functional group gave lower removal compared to ours. The other studies that removed higher than our study was based on using photodegradation or membranes which is extremely very expensive compared with our study